# Diversification of division mechanisms in endospore-forming bacteria revealed by analyses of peptidoglycan synthesis in *Clostridioides difficile*

Shailab Shrestha[1,2], Najwa Taib[3,4], Simonetta Gribaldo [3] & Aimee Shen [1] ✉

The bacterial enzymes FtsW and FtsI, encoded in the highly conserved *dcw* gene cluster, are considered to be universally essential for the synthesis of septal peptidoglycan (PG) during cell division. Here, we show that the pathogen *Clostridioides difficile* lacks a canonical FtsW/FtsI pair, and its *dcw*-encoded PG synthases have undergone a specialization to fulfill sporulation-specific roles, including synthesizing septal PG during the sporulation-specific mode of cell division. Although these enzymes are directly regulated by canonical divisome components during this process, *dcw*-encoded PG synthases and their divisome regulators are dispensable for cell division during normal growth. Instead, *C. difficile* uses a bifunctional class A penicillin-binding protein as the core divisome PG synthase, revealing a previously unreported role for this class of enzymes. Our findings support that the emergence of endosporulation in the *Firmicutes* phylum facilitated the functional repurposing of cell division factors. Moreover, they indicate that *C. difficile*, and likely other clostridia, assemble a distinct divisome that therefore may represent a unique target for therapeutic interventions.

Synthesis of cell wall peptidoglycan (PG) is essential for growth and division in most bacteria. The extra-cytoplasmic assembly of PG results from two sequential enzymatic reactions: a transglycosylation reaction that polymerizes the PG precursor Lipid II into glycan strands and a transpeptidation reaction that crosslinks these glycan strands together to form a protective meshwork[1].

Our understanding of this critical process has been transformed by the recent discovery that shape, elongation, division, and sporulation (SEDS) family proteins function as PG glycosyltransferases in complex with cognate class B penicillin-binding protein (bPBP) transpeptidases to synthesize PG[2–5]. Current models posit that specific SEDS-bPBP pairs function as the core PG synthases driving either cell elongation or division in rod-shaped bacteria[2,4–8]. These specialized

pairs of SEDS-bPBPs associate with specific multiprotein assemblies to mediate either cell elongation or cell division: lateral growth is typically driven by the SEDS-bPBP pair, RodA-MrdA, as a part of the elongasome, while septum formation is mediated by the SEDS-bPBP pair, FtsW-FtsI, as a part of the divisome.

Prior to the discovery of SEDS glycosyltransferases, class A penicillin-binding proteins (aPBPs) were the only known PG synthases with glycosyltransferase activity. Unlike monofunctional bPBP transpeptidases, aPBPs are bifunctional enzymes that harbor both glycosyltransferase and transpeptidase activities and they were presumed to be the primary PG synthases driving cell elongation and division[9]. However, recent evidence suggests that aPBPs often play non-essential, peripheral roles during these processes which is consistent with

[1]Department of Molecular Biology and Microbiology, Tufts University School of Medicine, Boston, MA, USA. [2]Program in Molecular Microbiology, Tufts University Graduate School of Biomedical Sciences, Boston, MA, USA. [3]Institut Pasteur, Université Paris Cité, Evolutionary Biology of the Microbial Cell Unit, Paris, France. [4]Institut Pasteur, Université Paris Cité, Bioinformatics and Biostatistics Hub, F-75015 Paris, France. ✉e-mail: aimee.shen@tufts.edu

their absence from the genomes of many obligate intracellular bacteria[3,10–12]. Indeed, aPBPs can function independently of the divisome and elongasome and appear to mainly modify and repair PG synthesized by SEDS-bPBP enzymes[9,12,13].

Notably, these models of bacterial PG synthesis primarily derive from studies in *Escherichia coli* and *Bacillus subtilis*, but studies in organisms with different cell morphologies and mechanisms of growth have variably supported and challenged these general models. For instance, all Actinobacteria and some Proteobacteria follow a polar growth model where cell growth is driven by PG synthesis at cell poles. In these organisms, cell elongation is largely mediated by aPBP activity[14–17]. Given the diversity of mechanisms involved in bacterial PG synthesis and the importance of studying these processes in diverse bacteria, the functional characterization of major PG synthases in different bacteria is an area of significant interest.

Although diverse mechanisms of cell elongation have been described in bacteria, cell division mechanisms appear to be broadly conserved. Divisome-guided septal PG synthesis is mediated by the essential SEDS-bPBP enzyme pair, FtsW-FtsI, which is considered to be universally conserved in all bacteria[5]. Genes encoding these enzymes are typically located within the division and cell wall (*dcw*) cluster[18], which contains numerous genes involved in PG synthesis and cell division. Recent phylogenetic analyses have revealed that the *dcw* locus is widely conserved across almost all bacterial phyla and likely originated in the Last Common Bacterial Ancestor billions of years ago[18].

Despite this extreme conservation, the SEDS gene in the *dcw* cluster of the model endospore-forming bacterium *Bacillus subtilis* does not encode FtsW, but rather SpoVE, a sporulation-specific SEDS glycosyltransferase that is critical for endospore formation[19–21]. In *B. subtilis*, SpoVE forms a complex with the sporulation-specific bPBP SpoVD to synthesize a thick protective PG layer known as the cortex during sporulation[22–24]. Notably, *B. subtilis* is a member of the Firmicutes phylum, which is the only bacterial phyla to have evolved endospore formation. Since the formation of highly resistant, mature spores depends on a series of PG transformations unique to endospore-forming bacteria, we hypothesized that the Firmicutes would exhibit greater variation in the composition and function of bPBP and SEDS genes in their *dcw* loci. In this work, we analyzed the distribution of PG synthase-encoding genes in the *dcw* loci of Firmicutes organisms and analyzed their function in the medically important clostridial pathogen, *Clostridioides difficile*. Here, we reveal a previously unappreciated diversification in the mechanisms by which *C. difficile*, and likely other clostridia, mediates cell division during vegetative growth and endospore formation.

## Results

### Diversity of *dcw*-encoded PG synthases based on sporulation potential

Firmicutes are unique among bacteria in forming endospores. While the last common ancestor of the Firmicutes is thought to have been an endospore former, the ability to sporulate has been independently lost among members of this phylum[25] (Fig. 1, Supplementary Fig. 1, Supplementary Data 2, 3). To analyze the distribution of genes encoding SEDS glycosyltransferases and bPBPs in the *dcw* cluster of both spore-forming and non-spore-forming Firmicutes organisms, we constructed a custom database consisting of 494 Firmicutes genomes representative of the six major classes described in this phylum. Next, we inferred the organism's ability to form endospores by determining the presence of *spoOA* and *spoIIE* in all Firmicutes genomes. These genes are both part of a core genomic signature of sporulating bacteria[26] and encode key regulators required for initiating and committing cells to sporulation, respectively[27,28], so their co-occurrence strongly suggests that a given species is a spore former. We searched for SEDS and bPBP homologs in all Firmicutes genomes and identified genes encoding

SEDS and bPBP enzymes in the *dcw* cluster based on their synteny with other genes of the *dcw* cluster (*mraW*, *mraY*, *ftsZ*, *mraZ*, and *murCDEF*). We also identified a pair of SEDS and bPBP homologs, RodA and MrdA, respectively, involved in cell elongation and located in a different locus. Finally, we identified SEDS homologs encoded adjacent to the gene encoding pyruvate carboxylase, PycA (Fig. 1, Supplementary Figs. 1, 2, Supplementary Data 2).

In agreement with what has been described in *Bacillus subtilis*[22], sporulating taxa from Bacilli encode two homologs of bPBPs in the *dcw* cluster, which most probably correspond to the canonical cell division PG synthesizing enzyme, FtsI, and the sporulation-specific PG synthesizing enzymes, SpoVD (Fig. 1). Non-sporulating Bacilli only have a single bPBP gene in their *dcw* clusters, suggesting that *spoVD* was lost from this locus coincident with the loss of sporulation; the remaining bPBP gene in the *dcw* locus presumably encodes *ftsI*. In contrast, sporulating Bacilli encode only a single SEDS glycosyltransferase gene in their *dcw* loci. This gene likely encodes SpoVE based on functional analyses in *B. subtilis*[19–21] and our finding that non-sporulating Bacilli lack SEDS genes altogether from their *dcw* loci (Fig. 1). Thus, SEDS genes appear to have been lost from the *dcw* cluster coincident with the loss of sporulation. Notably, all Bacilli carry a SEDS glycosyltransferase gene outside of the *dcw* cluster (usually adjacent to *pycA* (Fig. 1)) that codes for a canonical cell division FtsW ortholog; the essential function of this conserved gene has been validated in several members of the Bacilli[5,8,29,30], suggesting that *dcw*-encoded PG synthesizing enzymes became specialized to function exclusively during spore formation.

The same arrangement between sporulating and non-sporulating species can be observed among members of the Negativicutes and Limnochordia (Fig. 1), suggesting that the *dcw*-encoded SEDS and bPBP homologs present in the sporulating members of these Classes also encode SpoVE and SpoVD. Together, these analyses reveal that the presence of SEDS and bPBP genes in the *dcw* cluster is highly correlated with sporulation (Supplementary Data 3), and that the genes present in the *dcw* cluster of sporulating Firmicutes likely code for SpoVE and SpoVD, respectively, rather than the canonical cell division proteins FtsW and FtsI. Exceptions are represented by some members of the Clostridia, Tissierellia, and Erysipelotrichia, which display a mixed pattern (Supplementary Fig. 1, Supplementary Data 2). Surprisingly, many members of the Clostridia do not appear to have any extra copy of genes encoding division-specific SEDS and bPBP enzymes in their *dcw* clusters or elsewhere in the genomes, suggesting they completely lack the canonical cell division pair FtsW-FtsI. Therefore, we sought to define the functions of *dcw*-encoded SEDS and bPBP genes of the genetically tractable clostridial species *C. difficile*.

### *dcw*-encoded PG synthases control *C. difficile* septal PG synthesis during sporulation but not vegetative growth

To this end, we determined the effect of deleting the *dcw*-encoded SEDS and bPBP genes, *spoVE* and *spoVD*, on *C. difficile* growth and sporulation. Consistent with the results of a prior transposon screen indicating that *spoVE* and *spoVD* are non-essential for growth[31], we were able to create single deletion strains lacking *spoVD* or *spoVE*. These strains did not exhibit growth or morphological defects during vegetative growth (Fig. 2a, b), indicating that *C. difficile* SpoVD and SpoVE are not involved in vegetative cell division. Consistent with prior work[32,33], the *C. difficile* Δ*spoVD* strain failed to make heat-resistant spores (Supplementary Fig. 3) or synthesize a cortex layer based on transmission electron microscopy (TEM) analyses (Supplementary Fig. 4). Similar phenotypes were observed for the Δ*spoVE* strain, indicating that *C. difficile* SpoVD and SpoVE share similar functions with their orthologs in *B. subtilis* in mediating cortex synthesis[21–24]. Importantly, the sporulation defect of these mutants could be fully complemented from a chromosomal ectopic locus (Supplementary Fig. 3).

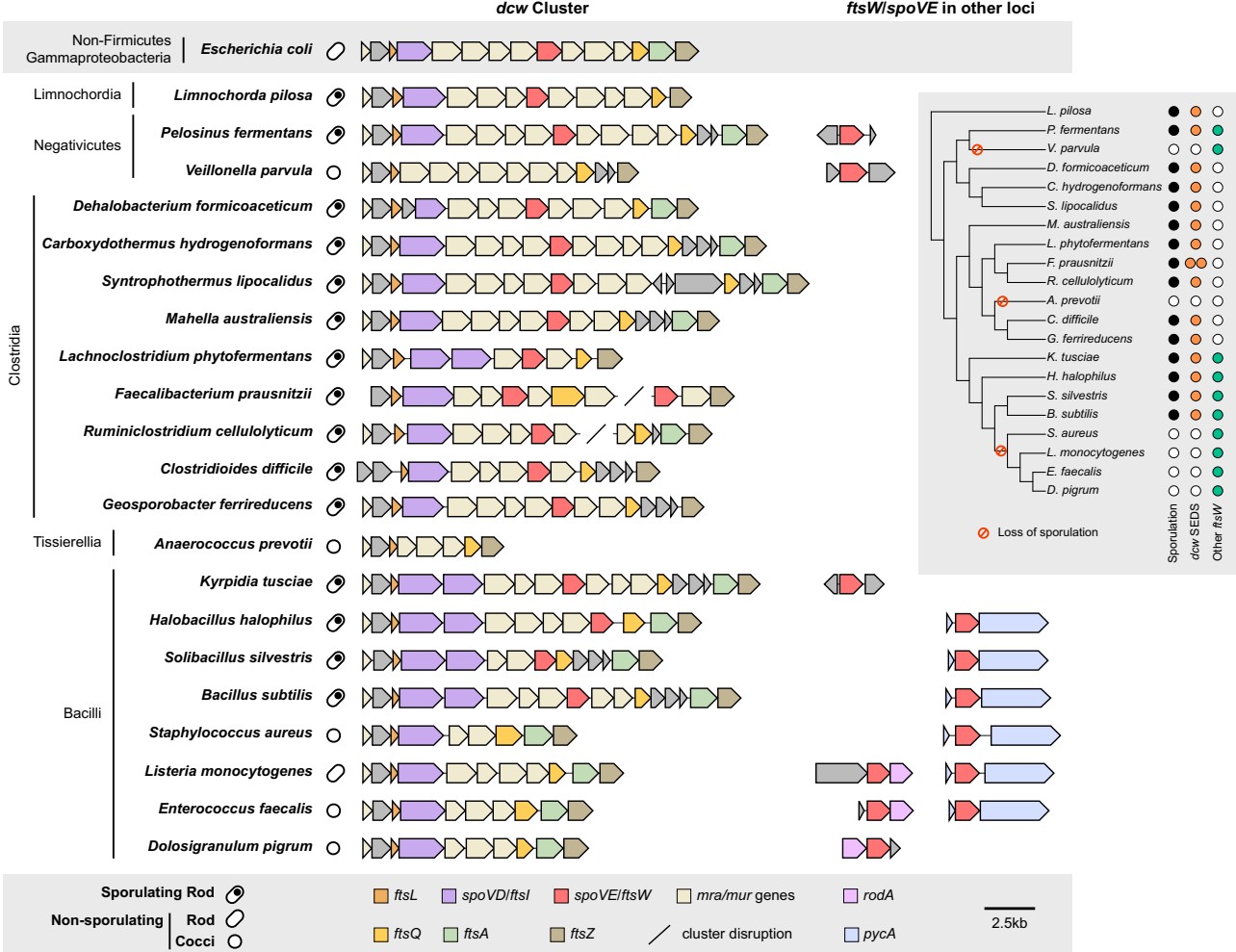

**Fig. 1 | Comparison of *dcw* cluster composition and occurrence of SEDS family proteins across diverse Firmicutes species.** Cell shape (rod or cocci) and sporulation ability are indicated. The ability to form spores was inferred by the presence of broadly conserved sporulation-specific genes *spoOA* and *spoIIE* in the genome[26]. All SEDS homologs that cluster with divisome-associated FtsW encoding genes are shown for each organism (Supplementary Fig. 2). For genome accession numbers, coordinates, and taxonomy information, see Supplementary Data 1. The inset tree shows phylogenetic relationships and the colored circles highlight the presence of sporulation genes (black), and *ftsW/spoVE* within (orange) and outside (green) the *dcw* cluster. For analyses of the full dataset, see Supplementary Fig. 1.

However, in contrast with *B. subtilis*, we observed that *C. difficile* Δ*spoVD* and Δ*spoVE* mutants appeared to sporulate at lower levels than wild-type (WT), with few cells exhibiting visible signs of sporulation during phase-contrast microscopy analyses (Supplementary Fig. S3). To investigate whether SpoVD and SpoVE affect sporulation processes earlier than cortex synthesis, we evaluated the ability of Δ*spoVD* and Δ*spoVE* cells to progress through the different morphological stages of sporulation via cytological profiling of sporulating cells[34,35]. These analyses revealed that most Δ*spoVD* and Δ*spoVE* cells showing morphological signs of sporulation were stalled at the asymmetric division stage, with a relatively small proportion completing engulfment compared to WT cells (Fig. 2c, Supplementary Fig. 5). Strikingly, the overall proportions of Δ*spoVD* and Δ*spoVE* cells exhibiting morphological signs of sporulation, i.e., cells that have completed or progressed beyond asymmetric division, were ~3-fold lower than WT (Fig. 2d). This effect at an earlier stage of sporulation is consistent with transcriptional analyses indicating that *C. difficile spoVD* and *spoVE* are expressed immediately at the onset of sporulation[36,37], in contrast with *B. subtilis spoVD* and *spoVE*, which are expressed in the mother-cell compartment after sporulating cells complete asymmetric division.

The decrease in apparent sporulation frequency in Δ*spoVD* and Δ*spoVE* mutants could be due to the enzymes regulating (1) asymmetric division through the synthesis of septal PG and/or (2) sporulation initiation via an unknown mechanism. To rule out the latter possibility, we compared the frequency of sporulation initiation in Δ*spoVD* and Δ*spoVE* cells relative to WT cells. Using a Spo0A-dependent P*spoIIE*::*mScarlet* transcriptional reporter, we found that Δ*spoVD* and Δ*spoVE* strains activate Spo0A, the master transcriptional regulator that initiates sporulation[27], at similar frequencies and levels relative to WT (Fig. 2e, f). In contrast, significantly fewer Δ*spoVD* and Δ*spoVE* cells compared to WT induced the expression of a reporter that is activated after the formation of the polar septum (P*sipL*::*mScarlet*) (Supplementary Fig. 6a)[36,37]. Consistent with the reporter data, RT-qPCR and western blot analyses of the Δ*spoVD* and Δ*spoVE* mutants confirmed that they activate Spo0A at WT levels but exhibit defects in activating later-acting sporulation-specific sigma factors that only become activated upon completion of asymmetric division (Supplementary Fig. 7, 8). Overall, our data suggest that *C. difficile* SpoVD and SpoVE play important roles in synthesizing septal PG during asymmetric division, in addition to their canonical function in synthesizing the spore cortex. Consistent with this model, we detected incomplete polar septum formation in Δ*spoVD* and Δ*spoVE*, but not WT, cells in transmission electron microscopy analyses (Fig. 2g). Moreover, in agreement with a previous study[33], *C. difficile* SpoVD localizes to polar septa during asymmetric division (Supplementary Fig. 9).

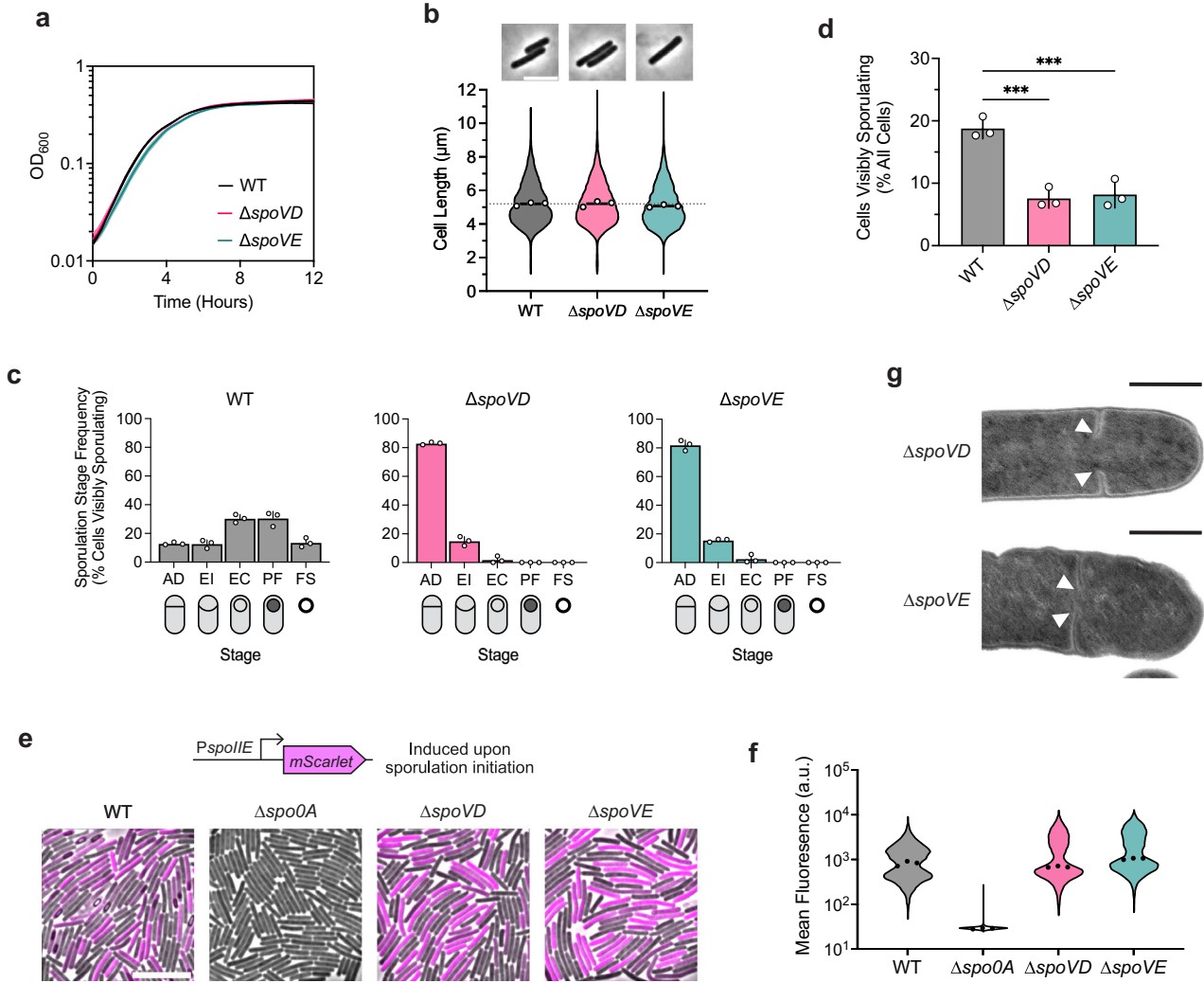

**Fig. 2 | The sporulation-specific PG synthases, SpoVD and SpoVE, are important for asymmetric, but not vegetative, division. a** Growth profiles of *C. difficile* wildtype (WT), Δ*spoVD*, and Δ*spoVE* strains in BHIS. Data are from a single experiment; mean and standard deviation curves are plotted from three biological replicates. **b** Violin plots showing length distributions and representative micrographs of cells sampled from BHIS cultures during mid-exponential growth (OD$_{600}$ -0.5). White circles indicate means from each replicate; black lines indicate average means; dotted line indicates the WT average mean for comparison across strains. Data from three biological replicates; >1500 cells per sample. Scale bar, 5 μm. **c**, **d** Cytological profiling of cells sampled from sporulation-inducing plates after 18–20 h of growth. Cells were assigned to five distinct stages based on their membrane (FM4-64) and DNA (Hoechst) staining and phase-contrast morphological phenotypes. For representative micrographs and stage assignment information, see Supplementary Fig. 5a. AD Asymmetric Division, EI Engulfment Initiated, EC Engulfment Completed, PF Phase bright/dark Forespore, FS Free Spore. White circles indicate means from each replicate, bars indicate average means and error

bars indicate standard deviation. ***$p$<0.001; statistical significance was determined using ordinary one-way ANOVA with Dunnett's test. Data from three independent experiments; >1000 total cells and >100 visibly sporulating cells per sample. Source data with exact *P* values are provided in the Source Data file. **e** Representative merged phase-contrast and fluorescence micrographs visualizing P$_{spoIIE}$::*mScarlet* transcriptional reporters in sporulating cells sampled from 70:30 plates after 14–16 h of growth. P$_{spoIIE}$ is induced upon sporulation initiation. The Δ*spo0A* strain serves as a negative control because it does not initiate sporulation. Scale bar, 10 μm. **f** Violin plots showing quantified mean fluorescence intensities of strains shown in **e**. Black dots represent median values from each replicate. Data from three independent experiments; >3000 cells per sample. **g** Transmission electron micrographs of Δ*spoVD* and Δ*spoVE* sporulating cells that fail to complete septum formation 24 h after sporulation induction (white arrows). Scale bars, 500 nm. >50 sporulating cells were analyzed per strain from two independent experiments for WT and Δ*spoVD* and one experiment for Δ*spoVE*.

## Conserved divisome components regulate cell division during spore formation but not vegetative growth in *C. difficile*

Septal PG synthesis requires the coordinated assembly and localization of numerous divisome components at the division site. If SpoVD and SpoVE mediate septal PG synthesis during asymmetric division, we reasoned that their activities are likely regulated by components of the divisome. Previous studies have implicated the widely conserved divisome sub-complex composed of FtsL, FtsQ, and FtsB (also known as FtsL, DivIB, and DivIC in some Firmicutes) in directly regulating the activity and localization of FtsW-FtsI[38–42]. Intriguingly, although *C. difficile* appears to lack functional FtsW and FtsI orthologs, it encodes

orthologs of their regulators (Supplementary Fig. 10). *cd630_26570* and *cd630_26500* are both located in conserved locations within the *dcw* cluster (Fig. 1) and encode putative membrane proteins with homology to FtsL and FtsQ, respectively. We also identified *cd630_34920* as encoding an FtsB homolog. We refer to these genes as *ftsL* (*cd630_26570*), *ftsQ* (*cd630_26500*), and *ftsB* (*cd630_34920*) from hereon (Supplementary Fig. 10).

Although *ftsL*, *ftsQ*, and *ftsB* encode proteins that are typically considered essential components of the divisome, a previous transposon screen in *C. difficile* identified these genes as non-essential but important for spore formation[31]. Consistent with this finding, we

readily obtained Δ*ftsL*, Δ*ftsQ*, and Δ*ftsB* deletion strains. These mutants showed no significant growth or morphological defects (Fig. 3a, b), indicating that FtsL, FtsQ, and FtsB likely fulfill non-canonical, sporulation-specific roles in *C. difficile*. Indeed, all three mutants formed heat-resistant spores less efficiently than WT, with Δ*ftsL* and Δ*ftsB* showing ~100-fold defects and Δ*ftsQ* showing a modest ~2-fold decrease (Supplementary Fig. 3). These sporulation defects could be complemented by expressing wildtype copies of their respective genes from a chromosomal ectopic locus (Supplementary Fig. 3). The milder phenotypes observed for Δ*ftsQ* are consistent with prior observations that FtsQ is only conditionally essential or completely absent in some bacterial species[43–45]. Notably, phase-contrast microscopy of sporulating cells revealed that all three mutants form phase-bright spores, albeit infrequently, and TEM analysis confirmed that the mutants can synthesize cortex PG, unlike Δ*spoVD* and Δ*spoVE* cells (Supplementary Fig. 4a). However, similar to Δ*spoVD* and Δ*spoVE* strains, incomplete polar septa were detected in TEM analyses of Δ*ftsL*, Δ*ftsQ*, and Δ*ftsB* strains, suggesting that these proteins are important for completing asymmetric division (Supplementary Fig. 4b).

Consistent with this observation, cytological profiling revealed that sporulating Δ*ftsL*, *ftsQ*, and Δ*ftsB* cells complete and progress beyond asymmetric division at a significantly lower frequency compared to WT (Fig. 3c, d, Supplementary Fig. 5). A small proportion (~2%) of Δ*ftsL* and Δ*ftsB* cells were able to progress beyond engulfment to make mature spores (Fig. 3d). The phenotype for Δ*ftsQ* was slightly less severe than for Δ*ftsL* and Δ*ftsB*, with ~4% of Δ*ftsQ* cells making phase-bright (cortex-positive) spores (Fig. 3d). However, a higher proportion of Δ*ftsQ* sporulating cells remained stalled at asymmetric division, suggesting that *C. difficile* FtsQ shares similar functions with its *B. subtilis* ortholog, DivIB, in regulating PG transformations during engulfment[46]. Since the PG synthesizing activity of FtsW-FtsI depends on direct interactions between the enzymes and the ternary FtsLQB sub-complex in the bacteria studied to date, we tested whether SpoVD and SpoVE form a divisome-like complex with FtsL, FtsQ, and FtsB by probing pairwise interactions using bacterial two-hybrid assays. This assay confirmed that SpoVD and SpoVE interact, consistent with these enzymes forming a cognate SEDS-bPBP pair as previously shown in *B. subtilis*[24] (Fig. 3e, f) and that *C. difficile* FtsL, FtsQ, and FtsB likely form a ternary sub-complex similar to homologs in other bacteria[47–50]. Importantly, we observed that *C. difficile* SpoVD interacts with FtsL and FtsQ, suggesting that the ternary sub-complex likely directly regulates the activity of SpoVE-SpoVD. Taken together, these data strongly suggest that *C. difficile* assembles a distinct polar divisome that partially relies on the sporulation-specific SEDS-bPBP pair, SpoVE-SpoVD, to synthesize the polar septum during endospore formation.

## *C. difficile*'s sole essential SEDS-bPBP pair mediates elongation

Although septal PG synthesis by SpoVD and SpoVE is important for asymmetric division, *C. difficile* cells that lack these proteins are still able to synthesize polar septa, albeit at lower rates. It is likely that the PG synthases that mediate vegetative cell division also contribute to septal PG synthesis during asymmetric division. Since *C. difficile* appears to lack obvious orthologs of the widely conserved divisome-specific SEDS-bPBP pair, we considered the involvement of all major PG synthases encoded in the *C. difficile* genome. *C. difficile* has a relatively minimal set of PG synthases: a single class A PBP (PBP1), three class B PBPs (PBP2, PBP3, and SpoVD), two SEDS proteins (RodA and SpoVE), and one monofunctional glycosyltransferase (MGT) (Fig. 4a). Among these synthases, we considered PBP1, PBP2, and RodA to be the most likely candidates for enzymes that contribute to septal PG synthesis during medial division since genes encoding these proteins were previously identified as being essential for vegetative growth[31]. Consistent with this and the prior finding that loss of PBP3 results in a sporulation defect[32], we confirmed that single deletions of genes encoding PBP3 and MGT do not significantly alter the growth of

vegetative cells, although they induce a modest increase in cell length (Supplementary Fig. 11).

Although RodA and PBP2 are the sole essential SEDS-bPBP pair, the genomic location of their respective genes adjacent to the *mreBCD* operon, which encodes critical components of the elongasome, predicts that they mediate elongation based on analyses in other bacteria[2]. Furthermore, *C. difficile* RodA branches with SEDS enzymes implicated in mediating cell elongation in other organisms and in a separate group from *dcw*-encoded SEDS proteins (Supplementary Fig. 2a, b), consistent with recent work suggesting that the functional divergence of SEDS paralogs to roles in cell division or elongation predates the Last Bacterial Common Ancestor[18]. To experimentally confirm the roles of RodA and PBP2 in cell elongation, we used CRISPR interference (CRISPRi)[51] to knock down the expression of their respective genes. Consistent with their essentiality, individual knockdowns produced significant growth defects upon induction of the CRISPR system (Fig. 4b). CRISPRi knockdown of either *rodA* or *pbp2* resulted in lateral bulging, rounding, and frequent lysis of cells (Fig. 4c). These phenotypes are characteristic of cells defective in cell elongation as observed in other rod-shaped bacteria[30,52–55] and stand in contrast to the phenotypes of divisome component CRISPRi knock-downs, which induce filamentation due to defects in septum formation. Thus, RodA and PBP2 are the core PG synthases during cell elongation in *C. difficile*.

## *C. difficile*'s aPBP drives septal PG synthesis during vegetative division

These analyses left the aPBP, PBP1, as the primary candidate for synthesizing septal PG during vegetative division in *C. difficile*. Since aPBPs are capable of both polymerizing and cross-linking PG, PBP1 should be able to substitute for both enzymatic activities fulfilled by a divisome-associated SEDS- bPBP pair. Indeed, PBP1 was previously suggested to be essential for vegetative growth[31], and its depletion results in cell filamentation[51]. Knockdown of the gene encoding PBP1 validated these prior reports, as it resulted in severe growth defects and cell filamentation phenotypes characteristic of cells deficient in cell division (Fig. 5a, b). Moreover, subcellular localization of PBP1 during vegetative growth showed significant enrichment at division septa (Fig. 5e, f).

Chemical inhibition of PBP1 activity with the glycosyltransferase inhibitor moenomycin also produced filamentous cells, suggesting a crucial role of PBP1 glycosyltransferase activity in cell division (Fig. 5d). These data contradict a prior report[56], which suggested that *C. difficile* cells are intrinsically resistant to moenomycin. Although moenomycin also inhibits the catalytic activity of MGTs, sensitivity to moenomycin was unchanged in an MGT-null mutant, suggesting that the drastic phenotypes observed in moenomycin-treated cells are exclusively due to the inhibition of PBP1 activity (Supplementary Fig. 12a). Consistent with this, growth defects from moenomycin treatment were rescued by overproduction of PBP1 (Supplementary Fig. 12c). Furthermore, these analyses revealed that *C. difficile* is sensitive to PBP1 overproduction likely due to dysregulation of PBP1 activity.

Next, we considered whether components of the polar divisome described above contribute to vegetative cell division. We reasoned that if this was the case, mutants lacking these components should be hypersensitive to moenomycin treatment. We found that the susceptibility of Δ*spoVD*, Δ*spoVE*, Δ*ftsL*, Δ*divIB*, and Δ*divIC* cells to moenomycin is unchanged compared to WT (Supplementary Fig. 12a, b). Overall, these results are consistent with a non-canonical divisome composition in *C. difficile*, where PBP1 is the major septal PG synthase during vegetative cell division.

## Discussion

The *dcw* cluster is well-conserved across all known bacterial phyla and was likely encoded in the Last Bacterial Common Ancestor[18]. While the

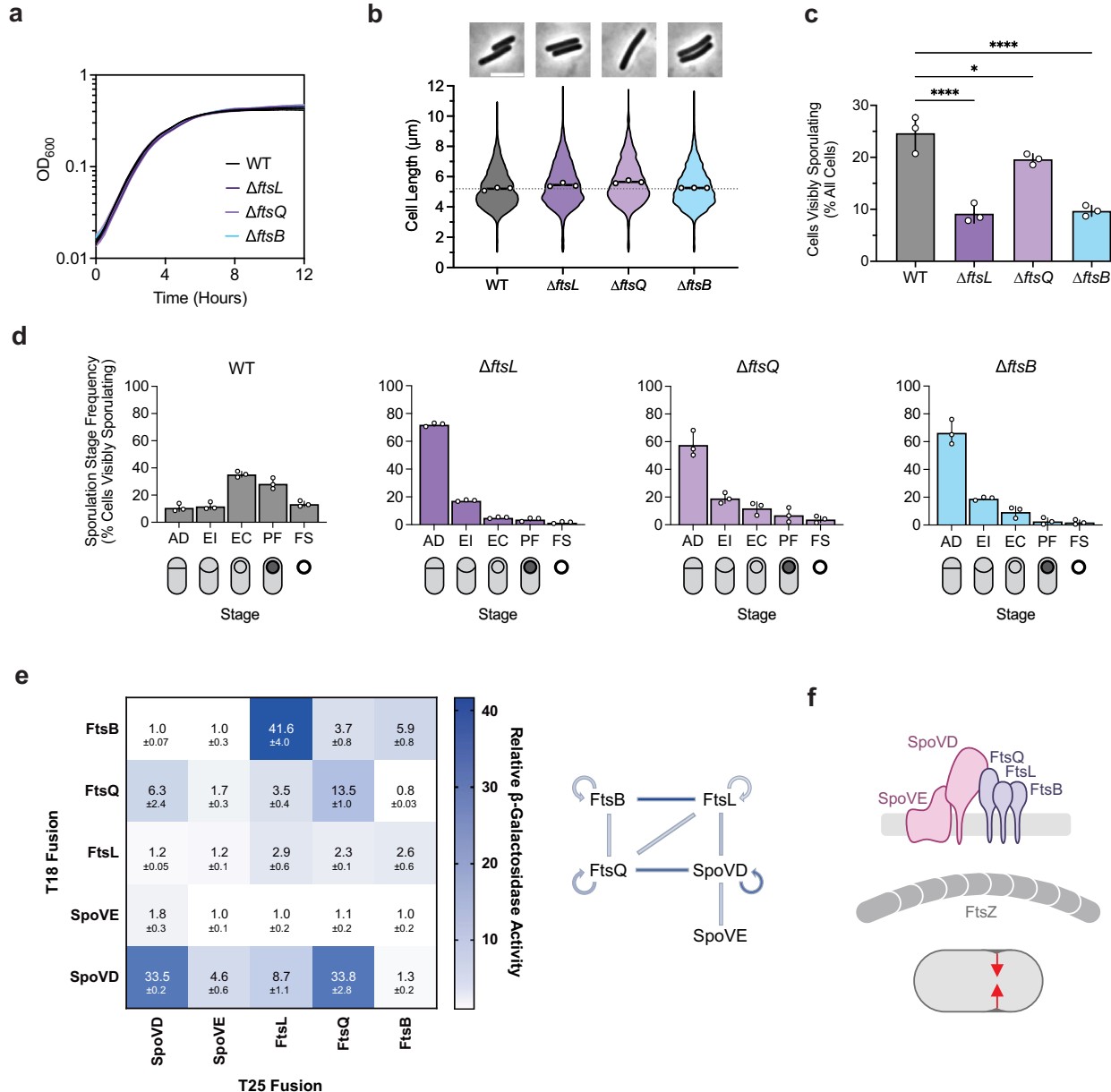

**Fig. 3 | The canonical divisome components, FtsL, FtsQ, and FtsB, are dispensable for vegetative cell division but are important for asymmetric division. a** Growth profiles of the indicated strains in BHIS. Data are from a single growth curve experiment; mean, and standard deviation curves are plotted from three biological replicates. **b** Violin plots showing cell length distributions and representative micrographs of cells sampled from BHIS cultures during exponential growth ($OD_{600}$ ~0.5). White circles indicate means from each replicate, black lines indicate average means, and the dotted line indicates the WT average mean for comparison across strains. Data from three biological replicates; >1500 cells per sample. Scale bar, 5 μm. **c, d** Cytological profiling of sporulating cells sampled from sporulation-inducing 70:30 plates after 18–20 h of growth. Cells were assigned to five distinct stages based on their membrane (FM4-64) and DNA (Hoechst) staining and their phase-contrast morphological phenotypes. For representative micrographs and stage assignment information, see Supplementary Fig. 5b. AD Asymmetric Division, EI Engulfment Initiated, EC Engulfment Completed, PF Phase-

bright/dark Forespore, FS Free Spore. White circles indicate means from each replicate, bars indicate average means and error bars indicate standard deviation. *$p < 0.05$, ***$p < 0.0001$; statistical significance was determined using an ordinary one-way ANOVA with Dunnett's test. Data from three independent experiments; >1000 total cells and >100 visibly sporulating cells per sample. Source data with exact $P$ values are provided in the Source Data file. **e** Bacterial two-hybrid analysis of interactions between components of the predicted polar divisome. The β-galactosidase activity was normalized to the negative control. Mean ± standard deviation from three biological replicates is indicated. The schematic shows interactions between different proteins where lines are colored according to the amount of β-galactosidase activity detected. **f** Schematic showing FtsL, FtsQ, and FtsB forming a divisome-like subcomplex with SpoVD and SpoVE. This polar divisome contributes to septal PG synthesis during asymmetric division. Created with BioRender.com.

order and composition of core *dcw* genes involved in PG synthesis and cell division have remained largely unchanged through billions of years of vertical inheritance, our analyses suggest that two key constituents of the cluster, *ftsW* and *ftsI*, have undergone a surprising functional divergence during the evolution of endospore formation. Rather than encoding the core PG synthases required for cell division

as they do in most bacteria, the SEDS and bPBP genes in the *dcw* cluster of sporulating Firmicutes likely encode enzymes specialized to function exclusively during endospore formation. This conclusion is supported by our finding that Firmicutes predicted to have lost the ability to sporulate also lack SEDS genes from their *dcw* cluster (Fig. 1, Supplementary Fig. 1, Supplementary Data 3). Furthermore, we show that,

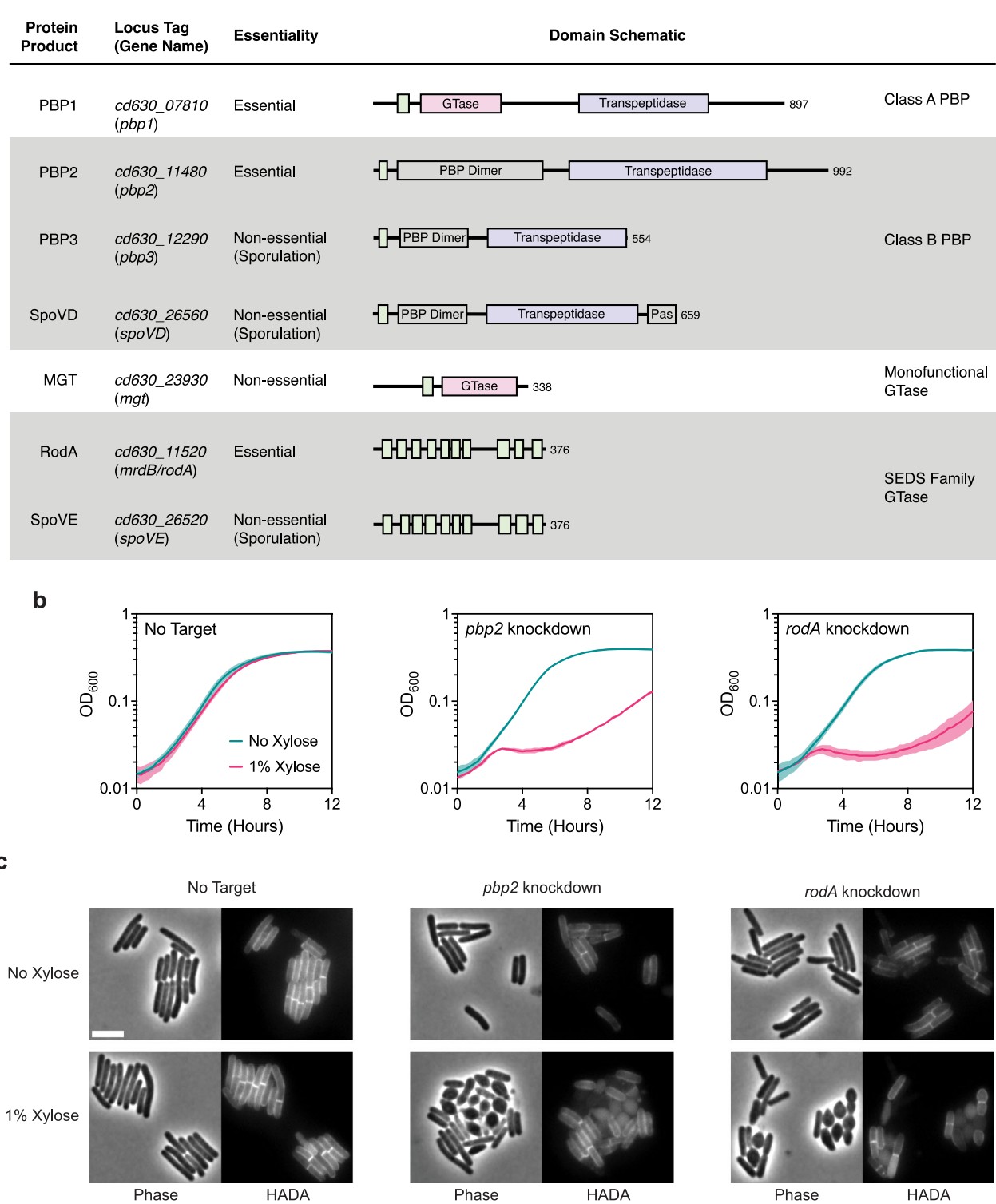

**Fig. 4 | RodA and PBP2 are involved in cell elongation. a** Major PG synthases encoded in the *C. difficile* strain 630 genome. Protein domains and catalytic sites predicted using HMMER are indicated in the schematic for each protein (hmmer.org). GTase, Glycosyltransferase Domain (Pink); PBP Dimer, PBP Dimerization Domain; Pas, PASTA Domain; Transpeptidase domain (Purple). Predicted transmembrane regions are depicted as green boxes. **b** Growth profiles of *pbp2* and *rodA* CRISPRi knockdown strains compared to a no target control strain. Data are from a single growth curve experiment; mean and standard deviation plotted from three biological replicates. **c** Representative micrographs showing morphological and PG incorporation phenotypes of control, *pbp2*, and *rodA* knockdown cells. Cells were grown in BHIS with or without xylose for 6–8 h. PG was labeled by incubation with a fluorescent D-amino acid (HADA). Scale bar, 5 μm. Data representative of multiple independent experiments.

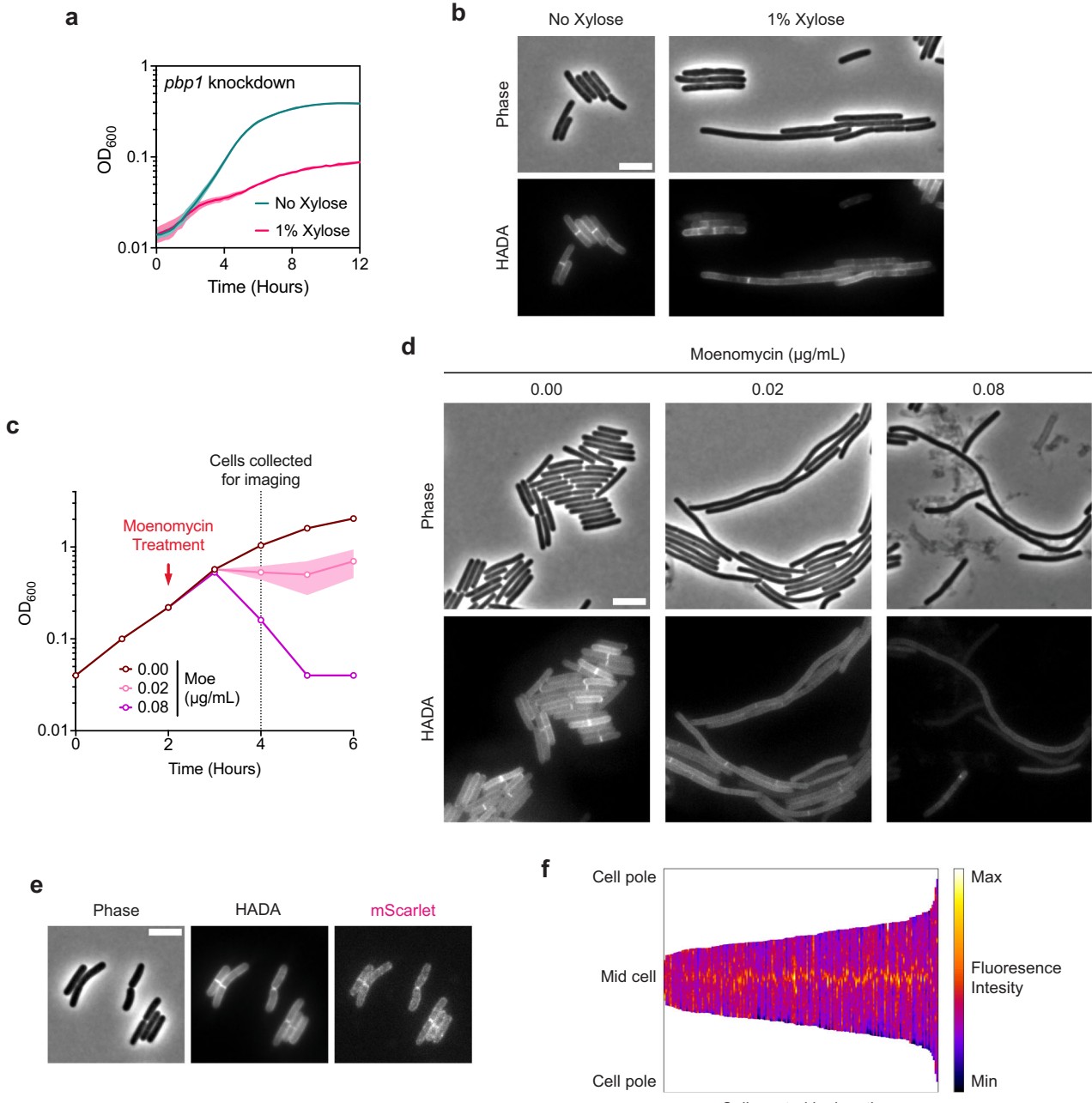

**Fig. 5 | The class A PBP, PBP1, is critical for cell division. a** Growth profile of the CRISPRi *pbp1* knockdown strain. Data from a single growth curve experiment; mean and standard deviation plotted from three biological replicates. **b** Representative micrographs showing morphological and PG incorporation phenotypes of *pbp1* knockdown cells. Cells were grown in BHIS with or without xylose for 6–8 h. PG was labeled by incubation with HADA. Scale bar, 5 μm. Data representative of multiple independent experiments. **c** Moenomycin treatment of WT cultures. Moenomycin was added after two hours of growth as indicated, and cells were collected and fixed for imaging 2 h after treatment. Mean and range are plotted from two biological replicates. **d** Representative micrographs showing morphological and PG incorporation phenotypes of moenomycin-treated WT cells. PG was labeled by

incubation with HADA. Scale bar, 5 μm. Data representative of multiple independent experiments. **e** Representative phase-contrast and fluorescence micrographs showing PG incorporation and subcellular localization of mScarlet-PBP1 in exponentially growing cells. PG was labeled by incubation with HADA, and *mScarlet-pbp1* was expressed using the native *pbp1* promoter from an ectopic locus. Scale bar, 5 μm. Similar patterns of localization were observed in multiple independent experiments. **f** Demograph showing fluorescence intensity across multiple cells representing subcellular localization of PBP1 in exponentially growing cells. Cells are aligned at the mid-cell and sorted by length. Data are from 173 cells from one experiment and are representative of two independent experiments.

like *B. subtilis*[23,24], the SEDS and bPBP genes in the *C. difficile dcw* cluster play sporulation-specific roles (Fig. 2).

However, in contrast with *B. subtilis*, where SpoVE and SpoVD function exclusively during cortex synthesis, we have identified a novel role for SpoVE and SpoVD in mediating septal PG synthesis during asymmetric division in *C. difficile*. We posit that this additional function

for SpoVE and SpoVD reflects their ancestral role in cell division. Since the last common ancestor of the Firmicutes is thought to have been a spore-former[25], it is likely that this ancestor diversified the function of its *dcw*-encoded cell division PG synthases, FtsW and FtsI, to include both asymmetric division and cortex formation. According to this evolutionary scenario, *C. difficile* SpoVE and SpoVD represent an

intermediate form that has retained its involvement in asymmetric division and cortex synthesis, while *B. subtilis* SpoVE and SpoVD became exclusively specialized for cortex synthesis. This interpretation is supported by our data suggesting that the PG synthesizing function of *C. difficile* SpoVD and SpoVE during asymmetric division is regulated by the divisome components FtsL, FtsQ, and FtsB (Fig. 3), which are essential regulators of FtsW and FtsI in other organisms, including *B. subtilis*[38–42].

Notably, the specialization of *dcw*-encoded SEDS and bPBP enzymes in mediating PG transformations during spore formation in the Firmicutes required that these organisms develop or repurpose other enzymes to drive vegetative cell division. Sporulating members of the Bacilli encode an additional bPBP in the *dcw* cluster adjacent to *spoVD* (Fig. 1), which appears to encode a functional ortholog of FtsI and was likely a product of a gene duplication event. Similarly, the evolution of a functional ortholog of FtsW encoded outside the *dcw* cluster in the Bacilli was likely facilitated by a gene duplication or horizontal gene transfer event. In contrast, many members of the Clostridia appear to lack *ftsW* and *ftsI* orthologs (Fig. 1), so a non-SEDS-bPBP PG synthase, namely the aPBP, PBP1, likely evolved to mediate cell division independent of the canonical cell division machinery.

While it is unclear why the function of SpoVE and SpoVD during asymmetric division is not conserved in *B. subtilis* and likely other Bacilli, it appears that their roles in this critical process will likely be observed only in spore formers that lack functional FtsW and FtsI orthologs. This is consistent with our observation that, unlike *C. difficile* and most other Clostridia, the Bacilli and Negativicutes appear to encode functional orthologs of FtsW and FtsI (Fig. 1), which likely mediate both medial and asymmetric division. Interestingly, the *B. subtilis* aPBP PBP1 is required for efficient septation during asymmetric division but not medial division[57], similar to how SpoVE-SpoVD are required for efficient asymmetric division in *C. difficile*. Given that the asymmetric division defects of *spoVD* and *spoVE* mutants in *C. difficile* are non-penetrant, it is likely that PBP1 is also a major driver of septal PG synthesis during sporulation. These observations suggest that the formation of the division septum during asymmetric division may have requirements for both aPBP and SEDS-bPBP activities, which may contribute to the differences between medial and polar septa observed in *B. subtilis*[58].

While our results strongly suggest that the essential PG synthases in *C. difficile*, PBP1 and RodA-PBP2, are functionally specialized to mediate cell division and elongation, respectively (Figs. 4, 5), it remains possible that these two distinct sets of PBPs participate in both processes. Indeed, PBP1 likely contributes to general cell wall synthesis and maintenance independent of its role in septation, given that aPBPs in other bacteria can function autonomously from the divisome[11]. This is supported by our observation of lower levels of HADA incorporation throughout the cell upon *pbp1* knockdown or moenomycin treatment (Fig. 5b, d). Furthermore, a recent study suggests that *C. difficile* PBP1 transpeptidase activity is likely non-essential[59], in which case, PBP2 may provide the missing transpeptidase activity. Indeed, redundancy in transpeptidase activities during cell division has been described in *B. subtilis* and *S. aureus* and may be more widely conserved than currently appreciated[8,60–62].

Crucially, whether and how *C. difficile* PBP1 activity is regulated during septal PG synthesis remains unclear. FtsL, FtsQ, and FtsB are dispensable during vegetative growth (Fig. 3), and C. *difficile* lacks homologs of FtsA and FtsN, which regulate divisome assembly and PG synthesis activity in many bacteria[1]. Thus, factors that substitute for the function of these widely conserved divisome proteins remain to be discovered. Interestingly, a cluster of three genes encoding mid cell-localizing proteins (MldABC) unique to *C. difficile* and closely related organisms appear to be important for cell division[63]. Since one of these proteins (MldA) contains a PG-binding SPOR domain which is commonly found in components of the divisome in other bacteria and was

recently implicated in regulating aPBP activity in *E. coli*[64], Mld proteins may be involved in regulating septal PG synthesis by PBP1. However, further study is required to define the components of the distinct divisome encoded by *C. difficile*.

Altogether, our findings provide novel insight into the evolution of PG-synthesizing enzymes in the Firmicutes and highlight the diversity of PG synthesis mechanisms employed by bacteria. These observations may explain some of the disparities in cell division mechanisms reported for model Firmicutes organisms such as *B. subtilis* and *S. aureus* compared to non-Firmicutes models such as *E. coli*, including differences in FtsW-FtsI enzyme dynamics and aPBP function in relation to the divisome[1,9]. Moreover, by revealing unique characteristics of *C. difficile*'s cell division machinery during sporulation and vegetative growth, these analyses may inform the development of more specific therapeutics against this important pathogen.

## Methods

### Gene synteny, homology searches, and phylogenetic analyses

Genomic loci containing *dcw* clusters shown in Fig. 1 were manually determined by searching for *dcw* genes (*mraZ*, *ftsW/spoVE*, *ftsZ*) and extracted from genomes in the Genbank database (accession numbers and coordinates are listed in Supplementary Data 1). Gene neighborhoods were visualized using Clinker[65] on the CAGECAT web server (https://cagecat.bioinformatics.nl/tools/clinker)[66].

For all other analyses presented in Supplementary Figs. 1 and 2 and Supplementary Data 3, we assembled a local databank of Firmicutes by selecting one proteome per genus as performed by ref. 67. Proteome selection was realized considering genome characteristics such as assembly level and category. The assembled databank contains 494 genomes listed in Supplementary Data 2. In order to build a reference phylogeny, exhaustive HMM-based homology searches (with the option --cut_ga) were carried out by using HMM profiles of bacterial ribosomal proteins from the Pfam 29.0 database as queries on the Firmicutes databank using the HMMER-3.1b2 package (hmmer.org)[68]. The conserved ribosomal proteins were aligned with MAFFT-v7.407 with the auto option and trimmed using BMGE-1.1[69]. The resulting trimmed alignments were concatenated into a supermatrix (497 taxa and 3776 amino acid positions). A maximum likelihood tree was generated using IQTREE-1.6.3[70] under the LG+I+G4 model with 1000 ultrafast bootstrap replicates.

Homology searches were performed using HMMSEARCH from the HMMER-3.1b2 package to screen all the proteomes in the Firmicutes databank for the presence of Spo0A, SpoIIE, SEDS glycosyltransferases, and bPBP homologs. We used the Pfam database to retrieve Pfam domains PF08769, PF07228, PF01098, and PF00905 for Spo0A, SpoIIE, SEDS glycosyltransferases, and bPBP respectively[71]. All hits were then manually curated using phylogeny and domain organization to discard false positives. For SEDS and bPBP sets of hits, the kept sequences were aligned with MAFFT v7.481 (--auto)[72], trimmed with trimAl 1.2rev59[73] and single gene trees were build using IQ-TREE v2.0.7[74] with best-fit model chosen according to BIC criterium.

We next used MacSyFinder2[75] to locate the hits in the genomes. We assessed the genes as located in the dcw cluster when at least four out of the genes *ftsW/spoVE*, *ftsI/spoVD*, *ftsA*, *mraW*, *mraY*, *ftsZ*, *mraZ*, and *murCDEF* cooccurred in the genome separated by no more than five other genes. The *rodA-mrdA* pair located out of the dcw cluster was assessed when they were found separated by no more than five other genes with the absence of *mraW* and *mraY* in their close synteny. Finally, we identified the pair *ftsW-pycA* when the two genes cooccurred in the genome, separated by no more than 3 other genes. The presence or absence of the studied proteins and their synteny in the genomes were mapped onto the trees using IToL[76].

The Jaccard similarity coefficients reported in Supplementary Data 3 were calculated using R and RStudio Version 1.4.1717 using the dataset from Supplementary Data 2. The Jaccard similarity coefficient

for each pairwise comparison was calculated by dividing the number of shared organisms by the total number of organisms in the two sets being compared.

## *C. difficile* strain construction and growth conditions

All *C. difficile* strains used in the study are derivatives of 630Δ*erm*. Mutant strains were constructed in a 630Δ*erm*Δ*pyrE* strain using *pyrE*-based allele-coupled exchange as previously described[77]. All strains used in the study are reported in Supplementary Data 4. *C. difficile* strains were grown from frozen glycerol stocks on brain heart infusion-supplemented (BHIS) agar plates with taurocholate (TA, 0.1% w/v). *C. difficile* strains harboring pIA33- or pRPF185-based plasmids were grown on media supplemented with thiamphenicol (10 μg/mL in liquid cultures and 5 μg/mL in agar plates). Cultures were grown at 37 °C under anaerobic conditions using a gas mixture containing 85% $N_2$, 5% $CO_2$, and 10% $H_2$.

## *E. coli* strain constructions

Supplementary Data 5 lists all plasmids used in the study, with links to plasmid maps containing all primer sequences used for cloning. Plasmids were cloned via Gibson assembly, and cloned plasmids were transformed into *E. coli* (DH5α or XL1-Blue strains). All plasmids were confirmed by sequencing the inserted region. Confirmed plasmids were transformed into the *E. coli* HB101 strain for conjugation with *C. difficile*. All *E. coli* HB101 strains used for conjugation are also listed in Supplementary Data 5.

## Plate-based sporulation assays

For assays requiring sporulating cells, cultures were grown to early stationary phase, back-diluted 1:50 into BHIS, and grown until they reached an $OD_{600}$ between 0.35 and 0.75. 120 μL of this culture was spread onto 70:30 (70% SMC media and 30% BHIS media) or SMC agar plates as indicated (40 ml media per plate). Sporulating cells were scraped from the plate and collected into phosphate-buffered saline (PBS), and sporulation levels were visualized by phase-contrast microscopy as previously described[78].

## Heat resistance assay

Heat-resistant spore formation was measured 18–22 h after sporulation induction on 70:30 agar plates by resuspending a sample of sporulating cells in PBS, dividing the sample into two, heat-treating one of the samples at 60 °C for 30 min, and comparing the colony forming units (CFUs) in the untreated sample to the heat-treated sample[79]. Heat-resistance efficiencies represent the average ratio of heat-resistant CFUs to total CFUs for a given strain relative to the average ratio for the wild-type strain.

## Western blot analysis

Samples were collected 17 h after sporulation induction on 70:30 agar plates and processed for immunoblotting. Sample processing involved multiple freeze-thaws in PBS followed by the addition of EBB buffer (9 M urea, 2 M thiourea, 4% SDS, 2 mM β-mercaptoethanol), boiling, pelleting, resuspension, and boiling again prior to loading on a gel. σ^F, σ^E, and Spo0A were resolved using 15% SDS–polyacrylamide gel electrophoresis (SDS–PAGE) gels, whereas SpoVD and SpoIVA were resolved using 12% SDS–PAGE gels. Proteins were transferred to polyvinylidene difluoride membranes, which were subsequently probed with rabbit (anti-SigF[36], anti-SigE[36], and anti-SpoVD; all at 1:1000 dilution) and mouse (anti-Spo0A[36] (1:1000 dilution) and anti-SpoIVA[36] (1:3000 dilution)) polyclonal primary antibodies, and anti-rabbit IR800 and anti-mouse IR680 secondary antibodies (LI-COR Biosciences, 1:20,000 dilution). Blots were imaged using a LiCor Odyssey CLx imaging system. The results shown are representative of multiple experiments.

## RT–qPCR analysis

RNA was harvested from sporulating cells 10–11 h after sporulating induction on 70:30 agar plates. Total RNA was processed according to published protocols[36]. Briefly, RNA was harvested using FastRNA Pro Blue Kit; DNA was removed by multiple DNAse I treatments, and mRNA was enriched using a MICROBExpress Bacterial mRNA Enrichment Kit for mRNA enrichment. Transcript levels were determined from cDNA templates prepared from a single experiment with three biological replicates per sample. Gene-specific primer pairs are provided in Supplementary Data 6. RT–qPCR was performed, using iTaq Universal SYBR Green supermix (BioRad), 50 nM of gene-specific primers, and an Mx3005P qPCR system (Stratagene) in a total volume of 25 μL[80]. The following cycling conditions were used: 95 °C for 2 min, followed by 40 cycles of 95 °C for 15 s and 60 °C for 1 min Statistical tests were performed using GraphPad Prism (GraphPad Software, San Diego, CA, USA).

## Bacterial two-hybrid analyses

Bacterial adenylate cyclase two-hybrid (BACTH) assays were performed using E. coli BTH101 cells as previously described[81]. BTH101 cells were freshly transformed with 100 ng of each BACTH assay plasmid and plated on fresh LB agar plates supplemented with 50 μg/ml kanamycin, 100 μg/ml Ampicillin, and 0.5 mM isopropyl β-D-thiogalactopyranoside (IPTG). Plates were incubated for 64–68 h at 30 °C, and β-galactosidase activity was quantified in Miller units as previously detailed[82]. The β-galactosidase activity of cells transformed with the empty pUT18C and pKT25 vectors was used as a negative control for normalization.

## Growth curve assays

For each replicate of the growth assay shown in Fig. 5c, a mid-log culture was normalized to a starting $OD_{600}$ ~0.05. After two hours of growth, the culture was aliquoted into three separate tubes and supplemented with moenomycin as indicated. The $OD_{600}$ value of each culture was measured every hour using a portable colorimeter (Biochrom WPA CO7500). For all other growth assays, stationary phase cultures were back-diluted 1:50 in BHIS and grown to mid-log phase ($OD_{600}$ 0.5). Log-phase cultures were normalized to a starting $OD_{600}$ ~0.01 and distributed into wells (150 μL per well) of a 96-well plate. The plate was incubated in an Epoch microplate spectrophotometer (Agilent BioTek) at 37 °C with linear shaking for 2 min prior to each time point. The $OD_{600}$ value for each well was recorded every 15 min. For growth assays involving strains with pIA33- or pRPF185-based plasmids, media was supplemented with 10 μg/mL thiamphenicol and either xylose or anhydrotetracycline as indicated. For assays involving moenomycin treatment, cultures were supplemented with different concentrations of moenomycin as indicated.

## Nucleoid, membrane, and cell wall labeling

Fluorescence microscopy was performed on sporulating cells using Hoechst 33342 (Molecular Probes; 15 μg ml⁻¹) and FM4-64 (Invitrogen; 1 μg ml⁻¹) to stain nucleoid and membrane, respectively. For cell wall labeling, HADA (Tocris Bioscience) was added to exponentially growing cell culture to a final concentration of 100–200 μM and incubated for ~2 min before cell fixation.

## Cell fixation

Cells were fixed by taking an 800 μL of culture in BHIS media and adding 200 μL of a 5× fixation solution containing paraformaldehyde and NaPO₄ buffer[83]. Samples were mixed and incubated in the dark for 30 min at room temperature, followed by 30 min on ice. Fixed cells were washed three times in PBS and resuspended in an appropriate volume of PBS depending on cellular density. Cells were imaged within 72 h after fixation.

## Microscope hardware

All samples for a given experiment were imaged from a single agar pad (1.5% low-melting point agarose in PBS). Phase-contrast micrographs shown in Supplementary Fig. 3 were acquired using a Zeiss Axioskop upright microscope with a 100× Plan-NEOFLUAR oil-immersion phase-contrast objective and a Hamamatsu C4742-95 Orca 100 CCD Camera. All other phase-contrast and fluorescence micrographs were obtained using a Leica DMi8 inverted microscope equipped with a 63× 1.4 NA Plan Apochromat oil-immersion phase-contrast objective, a high precision motorized stage (Pecon), and an incubator (Pecon) set at 37 °C. Excitation light was generated by a Lumencor Spectra-X multi-LED light source with integrated excitation filters. An XLED-QP quadruple-band dichroic beam-splitter (Leica) was used (transmission: 415, 470, 570, and 660 nm) with an external filter wheel for all fluorescent channels. FM4-464 was excited at 550/38 nm, and emitted light was filtered using a 705/72-nm emission filter (Leica); HADA and Hoechst were excited at 395/40, and emitted light was filtered using a 440/40-nm emission filter (Leica); mScarlet was excited at 550/38 nm, and emitted light was filtered using a 590/50-nm emission filter (Leica). Emitted and transmitted light was detected using a Leica DFC 9000 GTC sCMOS camera. 1 to 2 μm z-stacks were taken when needed with 0.21 μm z-slices.

## Microscopy image analyses

Images were acquired and exported using the LASX software. To avoid bleed-through of fluorescent signal into neighboring cells, a background subtraction method (Leica Instant Computational Clearing) was applied to the fluorescence images used for fluorescent reporter analyses shown in Fig. 2e, f and Supplementary Fig. 5. All other images were exported without further processing. After export, images were processed using Fiji[84] to remove out-of-focus regions via cropping. The best-focused Z-planes for all channels were manually selected to correct for any chromatic aberration. Image scaling was adjusted to improve brightness and contrast for display and was applied equally to all images shown in a single panel. For cell segmentation and quantification of length and fluorescent intensities, the MATLAB-based image analysis pipeline SuperSegger[85] was used with the default 60× settings. Data used for fluorescent reporter analyses shown in Fig. 2e, f and Supplementary Fig. 5 were filtered to remove spore particles by length and cells at image border by XY position. Visualization of quantified data and any associated statistical tests were performed using Prism 10 (GraphPad Software, San Diego, CA, USA).

Segmentation of cells and fluorescent intensity analyses used to generate data shown in Fig. 5f to analyze mScarlet-PBP1 localization were conducted using the MicrobeJ plugin[86] in ImageJ. Analyses of SpoVD-mScarlet localization and HADA incorporation in asymmetrically dividing cells shown in Supplementary Fig. 5 were conducted in ImageJ. Individual cells undergoing asymmetric division were isolated by manually identifying cells with a polar septum in the HADA channel. A 5-pixel wide pole-to-pole vector was drawn to gather fluorescent intensities across the cell. For each channel, the fluorescent intensity was normalized to the minimum and maximum values for each cell and plotted against the normalized cell length.

## Transmission electron microscopy

Sporulating cells were collected ~22 h after sporulation induction on 70:30 or SMC agar plates. Cells were fixed and processed for electron microscopy by the University of Vermont Microscopy Center[87]. Briefly, sporulating cultures were fixed in 2% glutaraldehyde, 2% paraformaldehyde in 0.1 M sodium cacodylate buffer, washed in 0.1 M cacodylate buffer, embedded in agarose, cross-linked with Karnovsky's buffer (1% paraformaldehyde, 2.5% glutaraldehyde in 0.1 M cacodylate buffer), washed in 0.1 M cacodylate buffer, and minced into 1 mm³ pieces. Samples were dehydrated through graded ethanols (35%, 50%, 70%, 85%, and 95%) and cleared twice in 100% propylene oxide. Samples were infiltrated with Spurr's epoxy resin (100%

polypropylene oxide) with increasing ratios, and embedded in Spurr's resin at 70 °C prior to sectioning. Semi-thin sections (1 μm) were cut on a Reichert Ultracut Microtome, stained with methylene blue-azure II, and evaluated for areas of interest. Ultra-thin sections (60–80 nm) were cut with a diamond knife, retrieved onto 200 mesh thin bar nickel grids, contrasted with uranyl acetate (2% in 50% ethanol) and Reynold's lead citrate. All TEM images were captured on a JEOL 1400 Transmission Electron Microscope (Jeol USA, Inc., Peabody, MA) with an AMT XR611 high-resolution 11-megapixel mid-mount CCD camera.

## Reporting summary

Further information on research design is available in the Nature Portfolio Reporting Summary linked to this article.

## Data availability

Data used to generate graphs, exact *P* values from statistical analyses, and uncropped scans of blots can be accessed in the Source Data file. All other data supporting the findings of this study are available within the article, Supplementary Data and Supplementary Information file. Any associated unprocessed data are available upon request. The Genbank and Pfam 29.0 databases were used in this study. Source data are provided with this paper.

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

## Acknowledgements

We acknowledge the University of Vermont Microscopy Imaging Core for processing the samples and acquiring the images for all Transmission Electron Microscopy analyses. We thank Craig D. Ellermeier,David S. Weiss, and Ute Müh for their input on the project and for sharing protocols and plasmids used in the study. We are grateful to members of the Shen lab for helpful discussions and feedback on the manuscript. The National Institute of Allergy and Infectious Diseases grant R01 AI122232 (to A.S.), and Burroughs Wellcome Fund for Investigators in Pathogenesis Award (to A.S.) provided funding for this work.

## Author contributions

S.S. and A.S. conceived the study. S.S. performed and analyzed experiments. S.S. and N.T. conducted bioinformatic analyses. A.S. supervised the study. S.S. and A.S. wrote the manuscript with input from N.T. and S.G. All authors reviewed and approved the manuscript.

## Competing interests

The authors declare no competing interests.
