## [Peer Review File · Nature Communications]

Diversification of division mechanisms in endospore-forming bacteria revealed by analyses of peptidoglycan synthesis in *Clostridioides difficile*Reviewer #1 (Remarks to the Author):

In their paper Shrestha and colleagues investigate cell wall synthesis during growth and sporulation in the firmicute *Clostridioides difficile* using bioinformatic and cell biological approaches. They find that *C. difficile* has evolved a remarkable specification of the cell wall synthesis genes. In particular the authors show that *C. difficile* lacks a canonical FtsI/FtsW pair that is in other bacteria required for septal PG synthesis. Instead a homolog of the bPBP was found in SpoVD and a single SEDS protein SpoVE was found in the dcw cluster of *C. difficile*. Deletion of SpoVD and SpoVE had no effect on vegetative growth, but cells were impaired in sporulation. Sporulation of the deletion strains was not completely abolished, but reduced and almost no heat resistant spores were formed. In a detailed analysis the authors provide cytological evidence for the stage of the interference with spore formation in the mutants. In other model bacteria, in which PG synthesis has been studied, the function of the FtsI/W pair is regulated by a complex composed of FtsL/DivIC/DivIB (firmicute nomenclature). In particular FtsL was found to be an essential part of the divisome. Therefore, it is highly surprising (but logical in the context described here), that FtsL and FtsQ (DivIB) can be deleted without any significant growth effect. In *C. difficile* deletion of FtsL and Q revealed a sporulation phenotype, indicating their function at the spore specific, asymmetric septum. Importantly, the authors can genetically complement the respective mutations. They go on and identify the class A PBP, PBP1, as the only important PG synthase during septum formation. They confirm this by localization studies, CRISPRi knock down and moenomycin sensitivity (which can be suppressed by PBP1 overexpression). In summary, the authors show in a very comprehensive study an unexpected setup for PG synthesis in *C. difficile*. Their work adds to the increasing knowledge about the diversity of bacterial cell biology. Even closely related species may employ variations of core systems to ensure their cell growth and differentiation. In this case the study was performed with an highly important pathogen and we are in need for more insights into their cell biology to adjust antibiotic treatment therapies. It becomes again clear, that there will be no one-solution-fits-all treatment for bacterial pathogens. I consider this study therefore, important and timely. I do not see major shortcomings. Congratulations to the authors for this great work. Below I list a few questions that remained open for me:

Questions

Did the authors test the sporulation efficiency in the PBP1 knock-down? To me it was unclear if the function of PBP1 was required on vegetative septa and maybe sporulation septa.

Figure S3 shows sporulation of different mutant strains. To judge best about sporulation efficiencies a membrane stain is used. Here phase contrast images are shown. Phase contrast imaging allows at best to judge later stages of sporulation only (not formation of the asymmetric septum etc.)

I really wonder if these findings are generalizable within the Clostridia, but I understand if showing this using any other member of this group will be for future work.

A catalytic mutant blocking the TP activity of PBP1 was described. It might be interesting to test this here, too – or did the authors try and it is not viable?

Reviewer #2 (Remarks to the Author):

The presented manuscript represents an interesting study on the synthesis of cell wall peptidoglycan during asymmetric cell division in sporulating *Clostridium difficile*. This work presents a wide range of mainly genetic, biochemical and imaging experiments with important controls included and also bioinformatics analysis. However, this study has a few issues, as mentioned mainly in major comments, that need to be solved before publication. Presently, the conclusion is not fully supported by the results.

Major comments

1. Fig. 2 and Supplementary Fig. 3 with the assignment of different stages of sporulation, I strongly recommend using already established and described stages of sporulation for Bacilli and Clostridia sporulation. I understand, that the authors wanted to discriminate different stages II but in Bacilli stage II is already divided into sub-stages i, ii and iii. This is also important because the

sporulation genes' nomenclature is according to these stages, for example, SpoIIE should be, according to the presented stages here, named SpoIE. The stages should be changed in the entire manuscript when appropriate.

2. Lines 186-188 - "Consistent with the reporter data, RT-qPCR and western blot analyses of the Δ spoVD and Δ spoVE mutants confirmed that they activate Spo0A at WT levels but exhibit defects in activating later-acting sporulation-specific sigma factors that only become activated upon completion of asymmetric division (Supplementary Fig. 7, 8)."

The western blot (Supplementary Fig. 8) shows the signals for Spo0A markedly lower in both Δ spoVD and Δ spoVE mutant strains. The authors need to comment on the discrepancies in these data.

3. Lines 172-174 - "This effect at an earlier stage of sporulation is consistent with transcriptional analyses indicating that *C. difficile* spoVD and spoVE are expressed immediately at the onset of sporulation (Fimlaid et al., 2013; Saujet et al., 2013)..." This statement is controversial, even though in Fig. 8. it is shown that in Δ spoVD strain no Spo0A is detected. It is crucial to show also that no Spo0A is present in Δ spoVE mutant strain because other transcriptional analysis (Pettit et al. BMC Genomics 2014, 15:160; Supp. Material - 12864_2013_5908_MOESM2_ESM.xlsx) showed that transcriptome log2fold change in 630 Δ erm Δ spo0A for spoVE is not significant.

4. This study claims that SpoVD and SpoVE are important for the formation of asymmetric septum and they show an example of incomplete septa in both Δ spoVD and Δ spoVE mutant strains by TEM. It would be important to show statistics of these incomplete septa since from fluorescence microscopy data it is not possible to distinguish complete and incomplete septa. It cannot be ruled out that these two proteins are important for the remodelling of sporulation septum and per se not its formation.

Minor comments

1. Supplementary Fig. 7. How you can explain such differences in statistical significance for wt and Δ spo0A in five different panels? Significance range from * $p < 0.05$ to *** $p < 0.001$ with what seem to be very similar measurements.

Reviewer #3 (Remarks to the Author):

The paper from Shrestha and co-workers describes an in-depth analysis of cell wall synthesis in *Clostridioides difficile* that reveals a surprising evolutionary twist on cell division that has broad implications for the underlying mechanism. In the well-studied model organisms, cell wall synthesis is carried out by two main classes of cell wall synthases: the class A PBPs (aPBPs) and complexes between SEDS proteins and class B PBPs (bPBPs). The SEDS-bPBP complexes RodA-PBP2 and FtsW-FtsI have been shown to be the essential synthesis proteins for cell elongation/shape and cell division, respectively, whereas the aPBPs are thought to fortify and repair the cell wall in support of the SEDS-bPBP systems. The results in this paper turn this model on its head with respect to cell division in *C. diff*.

Bioinformatic analysis revealed that the conserved dcw cluster in sporulating bacteria likely encodes a sporulation specific SEDS-bPBP system rather than the FtsW-FtsI division enzyme as in most non-sporulating bacteria. In many of these spore formers, like *B. subtilis*, an FtsW ortholog is encoded elsewhere in the genome. This was found not to be the case in *Clostridia*, suggesting that the sporulation specific SEDS-bPBP system might also work in division. However, in a series of beautiful experiments, the authors show that it is an aPBP, PBP1, that is an essential cell division enzyme in *C. diff* whereas the SEDS-bPBP system is used to form the spore septum where it is regulated by the conserved and canonical activation system that normally controls the FtsW-FtsI enzymes for medial division. The results reveal a surprising flexibility in the division mechanism, showing for the first time that it can be mediated by an aPBP as the sole essential cell wall synthase as opposed to FtsW-FtsI.

The paper is very well-written and clear. The experiments are all well-controlled and the conclusions are appropriate. It was a joy to read and review. I have no major concerns.

1) The legend for Fig. 1 was a little confusing. It states that RodA encoding genes were excluded from the analysis, yet there are rodA genes shown in the diagram? Also, in line 105 it is stated, "loss of sporulation (red genes)", but there are no red genes. Were the open circles supposed to be red?

2) Line 119: "...sporulating Firmicutes LIKELY code for..."

REVIEWER COMMENTS AND AUTHOR RESPONSES

Reviewer #1 (Remarks to the Author):

In their paper Shrestha and colleagues investigate cell wall synthesis during growth and sporulation in the firmicute *Clostridioides difficile* using bioinformatic and cell biological approaches. They find that *C. difficile* has evolved a remarkable specification of the cell wall synthesis genes. In particular the authors show that *C. difficile* lacks a canonical FtsI/FtsW pair that is in other bacteria required for septal PG synthesis. Instead a homolog of the bPBP was found in SpoVD and a single SEDS protein SpoVE was found in the dcw cluster of *C. difficile*. Deletion of SpoVD and SpoVE had no effect on vegetative growth, but cells were impaired in sporulation. Sporulation of the deletion strains was not completely abolished, but reduced and almost no heat-resistant spores were formed. In a detailed analysis the authors provide cytological evidence for the stage of the interference with spore formation in the mutants. In other model bacteria, in which PG synthesis has been studied, the function of the FtsI/W pair is regulated by a complex composed of FtsL/DivIC/DivIB (firmicute nomenclature). In particular, FtsL was found to be an essential part of the divisome. Therefore, it is highly surprising (but logical in the context described here), that FtsL and FtsQ (DivIB) can be deleted without any significant growth effect. In *C. difficile* deletion of FtsL and Q revealed a sporulation phenotype, indicating their function at the spore specific, asymmetric septum. Importantly, the authors can genetically complement the respective mutations. They go on and identify the class A PBP, PBP1, as the only important PG synthase during septum formation. They confirm this by localization studies, CRISPRi knock down and moenomycin sensitivity (which can be suppressed by PBP1 overexpression).

In summary, the authors show in a very comprehensive study an unexpected setup for PG synthesis in *C. difficile*. Their work adds to the increasing knowledge about the diversity of bacterial cell biology. Even closely related species may employ variations of core systems to ensure their cell growth and differentiation. In this case the study was performed with an highly important pathogen and we are in need for more insights into their cell biology to adjust antibiotic treatment therapies. It becomes again clear, that there will be no one-solution-fits-all treatment for bacterial pathogens. I consider this study therefore, important and timely. I do not see major shortcomings. Congratulations to the authors for this great work. Below I list a few questions that remained open for me:

Questions

We thank the reviewer for their positive comments and interesting questions, which are addressed below:

Did the authors test the sporulation efficiency in the PBP1 knock-down? To me it was unclear if the function of PBP1 was required on vegetative septa and maybe sporulation septa.

Although we posit that PBP1 likely mediates septal PG synthesis both during asymmetric division and medial division, we have not directly tested the effect of knocking down PBP1 on sporulation efficiency. Given that PBP1 knockdown affects cell growth and division, we anticipate that the results of such an experiment would be difficult to interpret because numerous rounds of growth and division of vegetative cells in sporulation-inducing media are needed before sporulation initiates (sporulation is induced ~9 hrs after cells are inoculated onto sporulation-inducing media). As a result, designing experiments with the proper controls to analyze PBP1 function during asymmetric division alone is likely to be challenging because PBP1 function cannot be fully ablated or the cells will not induce sporulation. However, if PBP1 function is not knocked-down sufficiently, PBP1 may be able to help a subset of cells complete asymmetric division. To address the Reviewer's important point, we now explicitly state our model of PBP1 function during asymmetric division with the following sentence in the Discussion section (lines 399-401):

“Given that the asymmetric division defects of spoVD and spoVE mutants in C. difficile are non-penetrant, it is likely that PBPI is also a major driver of septal PG synthesis during sporulation.”

Figure S3 shows sporulation of different mutant strains. To judge best about sporulation efficiencies a membrane stain is used. Here phase contrast images are shown. Phase-contrast imaging allows at best to judge later stages of sporulation only (not formation of the asymmetric septum, etc.)

Supplementary Fig. 3 shows phase-contrast images of sporulating samples corresponding with sporulation efficiencies for each strain. Here, sporulation efficiencies were determined through heat resistance assays that assess the ability of these strains to make heat-resistant mature spores, which are visible by phase-contrast microscopy. To assess the effect of different mutations throughout the process of sporulation, we carried out cytological profiling using membrane and DNA staining as reported in Fig. 2c-d, 3c-d, and Supplementary Fig. 5. We suspect that a mistake in Fig. 2 legend referring to Supplementary Fig. 3 for stage descriptions may have caused some confusion (the legend has been fixed to refer to Supplementary Fig. 5).

I really wonder if these findings are generalizable within the Clostridia, but I understand if showing this using any other member of this group will be for future work.

The Reviewer raises an excellent question. Given the lack of FtsW-I homologs in many Clostridial species, we suspect that our observations regarding SpoVD-SpoVE’s involvement in asymmetric division and PBPI’s involvement in medial division are likely to be conserved across many clostridial organisms. This is something we are currently pursuing for future study.

A catalytic mutant blocking the TP activity of PBPI was described. It might be interesting to test this here, too – or did the authors try and it is not viable?

We reference a previous study where the authors posit that PBPI TPase activity may be non-essential due to phenotypes observed through chemical inhibition of TPase activities in C. difficile (Sacco et al., 2022). However, neither we nor the authors of the referenced study (as far as we know) have tested a TPase catalytic mutant of PBPI, which would require a conditional knock-out system. While interesting, we believe this falls out of the scope of this work and is another area of interest we are pursuing for future study.

Reviewer #2 (Remarks to the Author):

The presented manuscript represents an interesting study on the synthesis of cell wall peptidoglycan during asymmetric cell division in sporulating *Clostridium difficile*. This work presents a wide range of mainly genetic, biochemical and imaging experiments with important controls included and also bioinformatics analysis. However, this study has a few issues, as mentioned mainly in major comments, that need to be solved before publication. Presently, the conclusion is not fully supported by the results.

Major comments

We thank the reviewer for their thorough evaluation of our work and their comments. Please find the point-by-point response to each issue raised below.

1. Fig. 2 and Supplementary Fig. 3 with the assignment of different stages of sporulation, I strongly recommend using already established and described stages of sporulation for Bacilli and Clostridia sporulation. I understand, that the authors wanted to discriminate different stages II but in Bacilli stage II

is already divided into sub-stages i, ii and iii. This is also important because the sporulation genes' nomenclature is according to these stages, for example, SpoIIE should be, according to the presented stages here, named SpoIE. The stages should be changed in the entire manuscript when appropriate.

*Although the stages of sporulation, as well as the nomenclature used for sporulation genes, are not always identical between *B. subtilis* and *C. difficile* (Pereira et al., 2013), we appreciate the push for consistency to avoid any confusion. We have changed the relevant figures and text to refer to the defined stages by specific designations (for instance, AD for Asymmetric Division, EI for Engulfment Initiated, etc.), which are described in the figure legends (detailed descriptions of each stage as identified by the cytological profiles of individual cells are still presented in Supplementary Fig. 5).*

2. Lines 186-188 - “Consistent with the reporter data, RT-qPCR and western blot analyses of the $\Delta spoVD$ and $\Delta spoVE$ mutants confirmed that they activate Spo0A at WT levels but exhibit defects in activating later-acting sporulation-specific sigma factors that only become activated upon completion of asymmetric division (Supplementary Fig. 7, 8).”

The western blot (Supplementary Fig. 8) shows the signals for Spo0A markedly lower in both $\Delta spoVD$ and $\Delta spoVE$ mutant strains. The authors need to comment on the discrepancies in these data.

*As Spo0A is present in vegetative cells, the onset of sporulation is initiated when phosphorylation enhances Spo0A activity (DNA binding), which leads to the expression of numerous early sporulation genes. Therefore, the levels of proteins encoded by genes expressed in a Spo0A-dependent manner are likely better indicators of Spo0A activation (as a proxy of sporulation initiation) than the levels of Spo0A itself. Our reporter analyses and qRT-PCR data are consistent with our interpretation of the western blot data, which shows that the levels of early factors (SigF and SpoVD) are similar between WT and the mutants ($\Delta spoVD$, $\Delta spoVE$, and \DeltaftsL), while the levels of later factors (SpoIVA and activated SigE) are lower in the mutants compared to WT. Regarding the noted discrepancy (lower levels of Spo0A observed in the mutants), we believe this may be explained by previous observations in *B. subtilis*, which show that Spo0A functions as a mother cell-specific transcription factor after completion of asymmetric division (Fujita and Losick, 2003; Marathe, 2020). Given that Spo0A regulates the expression of its respective gene, it is possible that the levels of Spo0A increase upon completion of asymmetric division. Furthermore, to avoid confusion, we have removed the following statement from the figure legend: “Spo0A levels reflect the level of sporulation induction (Fimlaid et al., 2013)” and replaced the previous Spo0A blot in Supplementary Fig. 8 with another where the same samples were run on a higher percentage gel. This allows us to discriminate between differentially phosphorylated forms of Spo0A.*

3. Lines 172-174 – “This effect at an earlier stage of sporulation is consistent with transcriptional analyses indicating that *C. difficile spoVD* and *spoVE* are expressed immediately at the onset of sporulation (Fimlaid et al., 2013; Saujet et al., 2013)...” This statement is controversial, even though in Fig. 8. it is shown that in $\Delta spoVD$ strain no Spo0A is detected. It is crucial to show also that no Spo0A is present in $\Delta spoVE$ mutant strain because other transcriptional analysis (Pettit et al. BMC Genomics 2014, 15:160; Supp. Material - 12864_2013_5908_MOESM2_ESM.xlsx) showed that transcriptome log2fold change in 630 $\Delta erm\Delta spo0A$ for *spoVE* is not significant.

*We are a little confused by the Reviewer's comment because Figure S8 in the original submission shows that Spo0A levels are present at WT or close-to WT levels in the $\Delta spoVD$ mutant; similar results were observed when we ran the same samples on a higher percentage gel to detect the differentially phosphorylated forms of Spo0A. With respect to the referenced statement being controversial, we note that two independent transcriptional analyses conducted on sporulating cell populations indicate that *spoVD* and *spoVE* are expressed early during sporulation in a likely Spo0A-dependent manner (Fimlaid et al., 2013; Saujet et al., 2013). In contrast, it is important to note that the other referenced study (Pettit*

et al., 2014) was conducted on exponentially growing cultures in rich media rather than cells grown in sporulation-inducing media (as in the other two studies). Since the number of cells that have induced sporulation is likely very low in this condition, false negatives are more likely to occur.

4. This study claims that SpoVD and SpoVE are important for the formation of asymmetric septum and they show an example of incomplete septa in both Δ spoVD and Δ spoVE mutant strains by TEM. It would be important to show statistics of these incomplete septa since from fluorescence microscopy data it is not possible to distinguish complete and incomplete septa. It cannot be ruled out that these two proteins are important for the remodelling of sporulation septum and per se not its formation.

*It is unclear what is being distinguished here in terms of “remodeling” and “formation,” but our cytological profiling data indicate that efficient completion of septum formation is dependent on SpoVD and SpoVE, since fewer Δ spoVD and Δ spoVE cells complete asymmetric division. It is important to note that the cytological profiling cannot detect partially completed polar septa (stage “a” in Supplementary Figure 5b) unless super-resolution methods or transmission electron microscopy are used. If the Reviewer’s distinction refers to the synthesis of the foundational septal layer vs. remodeling of this layer during septum formation, we believe that the experiments required to resolve this are out of the scope presented here. Taken together with evidence for involvement in foundational septal PG synthesis by divisome-associated homologs of SpoVD and SpoVE (FtsI and FtsW) in other bacteria, in our opinion, our data provide sufficient evidence to support our model that SpoVD and SpoVE in *C. difficile* are important drivers of septal PG synthesis during asymmetric division.*

Minor comments

1. Supplementary Fig. 7. How you can explain such differences in statistical significance for wt and Δ spo0A in five different panels? Significance range from * $p < 0.05$ to *** $p < 0.001$ with what seem to be very similar measurements.

Since the Y-axes are shown in logarithmic scale, the variation in the data, especially when comparing samples with smaller values to samples with larger values, might not be apparent. Nevertheless, to give the Reviewer more insight into the variation within the data, here are the values plotted on a linear scale, which might allow easier interpretation of the statistics (note the different ranges for the Y-axes):

Reviewer #3 (Remarks to the Author):

The paper from Shrestha and co-workers describes an in-depth analysis of cell wall synthesis in

Clostridioides difficile that reveals a surprising evolutionary twist on cell division that has broad implications for the underlying mechanism. In the well-studied model organisms, cell wall synthesis is carried out by two main classes of cell wall synthases: the class A PBPs (aPBPs) and complexes between SEDS proteins and class B PBPs (bPBPs). The SEDS-bPBP complexes RodA-PBP2 and FtsW-FtsI have been shown to be the essential synthesis proteins for cell elongation/shape and cell division, respectively, whereas the aPBPs are thought to fortify and repair the cell wall in support of the SEDS-bPBP systems. The results in this paper turn this model on its head with respect to cell division in *C. diff*.

Bioinformatic analysis revealed that the conserved *dcw* cluster in sporulating bacteria likely encodes a sporulation specific SEDS-bPBP system rather than the FtsW-FtsI division enzyme as in most non-sporulating bacteria. In many of these spore formers, like *B. subtilis*, an FtsW ortholog is encoded elsewhere in the genome. This was found not to be the case in Clostridia, suggesting that the sporulation specific SEDS-bPBP system might also work in division. However, in a series of beautiful experiments, the authors show that it is an aPBP, PBP1, that is an essential cell division enzyme in *C. diff* whereas the SEDS-bPBP system is used to form the spore septum where it is regulated by the conserved and canonical activation system that normally controls the FtsW-FtsI enzymes for medial division. The results reveal a surprising flexibility in the division mechanism, showing for the first time that it can be mediated by an aPBP as the sole essential cell wall synthase as opposed to FtsW-FtsI.

The paper is very well-written and clear. The experiments are all well-controlled and the conclusions are appropriate. It was a joy to read and review. I have no major concerns.

1) The legend for Fig. 1 was a little confusing. It states that RodA encoding genes were excluded from the analysis, yet there are rodA genes shown in the diagram? Also, in line 105 it is stated, “loss of sporulation (red genes)”, but there are no red genes. Were the open circles supposed to be red?

2) Line 119: “...sporulating Firmicutes LIKELY code for...”

We thank the reviewer for their positive comments and for pointing out these issues. We agree with both concerns and have changed the Fig. 1 legend (line 99-106) and the referred phrase (line 124) to address them.

References

- Fimlaid, K.A., Bond, J.P., Schutz, K.C., Putnam, E.E., Leung, J.M., Lawley, T.D., Shen, A., 2013. Global analysis of the sporulation pathway of *Clostridium difficile*. *PLoS Genet.* 9, e1003660. <https://doi.org/10.1371/journal.pgen.1003660>
- Fujita, M., Losick, R., 2003. The master regulator for entry into sporulation in *Bacillus subtilis* becomes a cell-specific transcription factor after asymmetric division. *Genes Dev.* 17, 1166–1174. <https://doi.org/10.1101/gad.1078303>
- Marathe, A., 2020. Molecular Insight into the Regulation of Cell Differentiation by a Master Regulator Spo0A in starving *Bacillus subtilis* cells. University of Houston.
- Pereira, F.C., Saujet, L., Tomé, A.R., Serrano, M., Monot, M., Couture-Tosi, E., Martin-Verstraete, I., Dupuy, B., Henriques, A.O., 2013. The spore differentiation pathway in the enteric pathogen *Clostridium difficile*. *PLoS Genet.* 9, e1003782. <https://doi.org/10.1371/journal.pgen.1003782>
- Pettit, L.J., Browne, H.P., Yu, L., Smits, W.K., Fagan, R.P., Barquist, L., Martin, M.J., Goulding, D., Duncan, S.H., Flint, H.J., Dougan, G., Choudhary, J.S., Lawley, T.D., 2014. Functional genomics reveals that *Clostridium difficile* Spo0A coordinates sporulation, virulence and metabolism. *BMC Genomics* 15, 160. <https://doi.org/10.1186/1471-2164-15-160>

- Sacco, M.D., Wang, S., Adapa, S.R., Zhang, X., Gongora, M.V., Gatdula, J.R., Lewandowski, E.M., Hammond, L.R., Townsend, J.A., Marty, M.T., Wang, J., Eswara, P.J., Jiang, R.H.Y., Sun, X., Chen, Y., 2022. A unique class of Zn²⁺-binding PBPs underlies cephalosporin resistance and sporogenesis of *Clostridioides difficile*. <https://doi.org/10.1101/2022.01.04.474981>
- Saujet, L., Pereira, F.C., Serrano, M., Soutourina, O., Monot, M., Shelyakin, P.V., Gelfand, M.S., Dupuy, B., Henriques, A.O., Martin-Verstraete, I., 2013. Genome-Wide Analysis of Cell Type-Specific Gene Transcription during Spore Formation in *Clostridium difficile*. *PLOS Genet.* 9, e1003756. <https://doi.org/10.1371/journal.pgen.1003756>